# Analyses of the Effect of Peptidoglycan on Photocatalytic Bactericidal Activity Using Different Growth Phases Cells of Gram-Positive Bacterium and Spheroplast Cells of Gram-Negative Bacterium

Akane Saikachi [1,†], Kotone Sugasawara [1,†] and Tomonori Suzuki [1,2,*]

[1] Department of Applied Biological Science, Tokyo University of Science,
2641, Yamazaki, Noda, Chiba 278-8510, Japan; 6419520@ed.tus.ac.jp (A.S.); 6420519@ed.tus.ac.jp (K.S.)

[2] Photocatalysis International Research Center, Tokyo University of Science,
2641, Yamazaki, Noda, Chiba 278-8510, Japan

* Correspondence: chijun@rs.tus.ac.jp

† They are contributed equally to this work.

**Abstract:** We conducted photocatalytic experiments focusing on the peptidoglycan layer to elucidate the details of the mechanism of photocatalytic sterilization. The previous study of our laboratory suggested that the presence of the peptidoglycan layer increases the bactericidal effect. To further verify it, the following experiments were performed: experiments on cells with different peptidoglycan layer thickness used *Lactobacillus plantarum* cells with different growth phases, experiments on cells with the thin peptidoglycan layer used *Escherichia coli* cells and spheroplast cells from which the peptidoglycan layer was removed from *E. coli* cells. The bactericidal effects increased as the growth progresses of *L. plantarum*. It was confirmed by TEM that the thickness of the peptidoglycan layer increased with cell growth. The survival rates of *E. coli* intact cells were significantly lower than those of spheroplast cells. These results strongly suggest that the peptidoglycan layer enhances the photocatalytic bactericidal effect. As a result of allowing the photocatalytic reaction to act on peptidoglycan, the amount of hydroxyl radical was smaller, and the amount of hydrogen peroxide was higher than in the absence of peptidoglycan. It is suggested that peptidoglycan may convert produced hydroxyl radical to hydrogen peroxide.

**Keywords:** photocatalysts; $TiO_2$; bactericidal effect; peptidoglycan; growth phase; spheroplast

## 1. Introduction

Recently, environmental pollution in water and air by microorganisms has come to be regarded as a problem, and among the problems, particular attention has been allocated to the influence of microorganisms on the human body such as nosocomial infections and infectious diseases caused by airborne bacteria [1–5]. The photocatalyst, which is inexpensive and has a low environmental impact, is attracting attention as a solution to these problems.

The photocatalyst is a substance that promotes a chemical change (oxidation-reduction reaction) of another substance by using the energy of light, and the most common one is $TiO_2$. When anatase-type $TiO_2$ is irradiated with UV light, reactive oxygen species (ROS) such as hydroxyl radicals are generated on the surface, and they decompose organic matters existing around the photocatalyst into water and carbon dioxide [6–10]. It is the photocatalytic sterilization that utilizes this reaction. So far, various studies have been conducted on the sterilization mechanism of photocatalysts [11–17] and its application [3,5,18–21]. However, there is a problem that the details of the sterilization mechanism by the photocatalytic reaction have not been clarified.

Although some studies have reported that higher bactericidal efficiency on Gram-negative bacteria than Gram-positive bacteria, these studies do not deeply study the effect of the peptidoglycan layer [22–25]. In addition, these studies do not consider the effects of spores resistant to physicochemical stress and the effects of extracellular secretory substances on photocatalytic sterilization. A study by Sunada et al. (2003) focused on the cell wall, the outer structure of the bacterial cells [26]. This study reported that ROS generated by the photocatalytic reaction acts on the outer membrane and the cell membrane of bacteria but does not act on the peptidoglycan layer and only passes through. In response to the study, a study in our laboratory was conducted with the aim of clarifying if ROS really does not act on the peptidoglycan layer and whether the bactericidal effect differs depending on the presence or absence of the peptidoglycan layer [27]. Using *Lactobacillus plantarum*, Gram-positive bacterium, the protoplast cells without the peptidoglycan layer were prepared, and the effects of the peptidoglycan layer on the photocatalytic bactericidal effect were analyzed by comparing the survival rate of the protoplast cells and the intact cells. As the results, the survival rate of the intact cells was significantly lower than the protoplast cells, it is suggested that the peptidoglycan layer enhances the bactericidal effect of the photocatalytic reaction.

Therefore, the purpose of this study is to clarify the mechanism of photocatalytic sterilization by multifaceted analyses of the effect of the peptidoglycan layer on the photocatalytic bactericidal effect. This study is structured as follows, (i) the effect of the photocatalytic sterilization on Gram-positive bacterium differing in the thickness of the peptidoglycan layer with the growth phase, (ii) the effect of the photocatalytic sterilization on spheroplast cell which is Gram-negative bacterium cell made by removing the peptidoglycan layer from the cell wall, (iii) the quantitative determination of hydroxyl radical and hydrogen peroxide generated when the peptidoglycan is photocatalytically treated.

## 2. Results

### 2.1. The Survival Rate of L. Plantarum as a Gram-Positive Bacterium on the Photocatalytic Reaction

2.1.1. The Survival Rates of *L. plantarum* JCM1142[T] in Each Growth Phase

To verify whether the bactericidal effect of the photocatalytic reaction differs depending on the thickness of the peptidoglycan layer, we performed the photocatalytic reaction with $TiO_2$-coated glass on the *L. plantarum* cells in each growth phase for 0, 20, 40, 60, and 120 min. The survival rates of the early log phase, the medium log phase, the late log phase, the early stationary phase cells decreased to about 54%, about 30%, about 28%, and about 16% after 40 min of the photocatalytic reaction, and about 20%, about 13%, about 7%, and about 8% after 1 h of the photocatalytic reaction, respectively (Figure 1). As a result of the significance test, it was shown that there were significant differences between the survival rates in the early log phase and the early stationary phase after 40 min ($p < 0.001$) and after 1 h ($p < 0.01$) of the photocatalytic reaction.

2.1.2. The Comparison of Survival Rates of *L. plantarum* JCM1142[T] with and without the Photocatalytic Reaction

The survival rates of *L. plantarum* cells under negative control conditions ([$TiO_2$ (−), UVA (−), early log phase], [$TiO_2$ (−), UVA (−), early stationary phase], [$TiO_2$ (−), UVA (+), early log phase], [$TiO_2$ (−), UVA (+), early stationary phase], [$TiO_2$(+), UVA (−), early log phase], and [$TiO_2$ (+), UVA (−), early stationary phase]) indicated about 86%, 96%, 80%, 98%, 89%, and 81% after 2 h of the photocatalytic reaction, respectively (Figure 2).

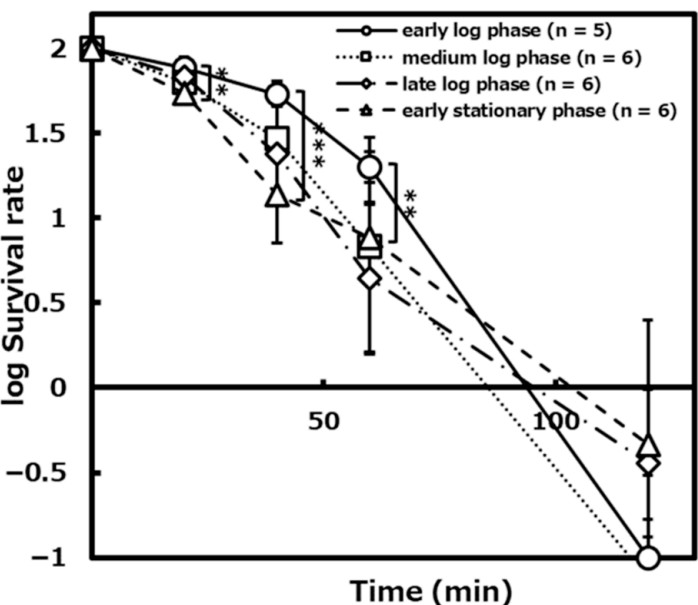

**Figure 1.** The survival rates of *L. plantarum* JCM1142[T] at each growth phase on the photocatalytic reaction with TiO$_2$-coated glass for 0, 20, 40, 60, and 120 min.([early log phase]: $n = 5$, [medium log phase]: $n = 6$, [late log phase]: $n = 6$, [early stationary phase]: $n = 5$). Error bars indicate the mean value $\pm$ standard deviation of experimental results repeated $n$ times. ([20 min]: $p < 0.01$, [40 min]: $p < 0.001$, [60 min]: $p < 0.01$). The double asterisk (**) and triple asterisk (***) indicate significant differences in the survival rates of the early log phase and early stationary phase at $p < 0.01$ and $p < 0.001$, respectively.

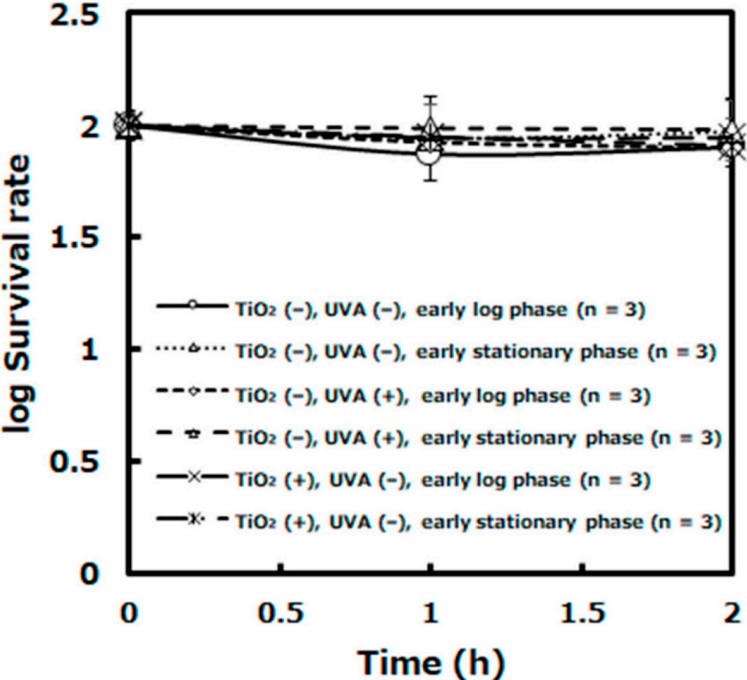

**Figure 2.** The survival rates of *L. plantarum* JCM1142[T] on the photocatalytic reaction under negative control conditions. ([TiO$_2$ (−), UVA (−), log early phase] [TiO$_2$ (−), UVA (−), stationary early phase]. [TiO$_2$ (−), UVA (+), log early phase] [TiO$_2$ (−), UVA (+), stationary early phase]. [TiO$_2$ (+), UVA (−), log early phase] [TiO$_2$ (+), UVA (−), stationary early phase]: $n = 3$). Error bars indicate the mean value $\pm$ standard deviation of experimental results repeated $n$ times.

### 2.2. TEM Observation of L. plantarum Cells in Each Growth Phase

In order to verify whether the results that *L. plantarum* JCM1142$^T$ cells on the photocatalytic reaction tended to be killed with cell growth were due to the thickening of the peptidoglycan layer with cell growth, the early log phase and the early stationary phase of *L. plantarum* cells were observed with a transmission electron microscope (Figure 3).

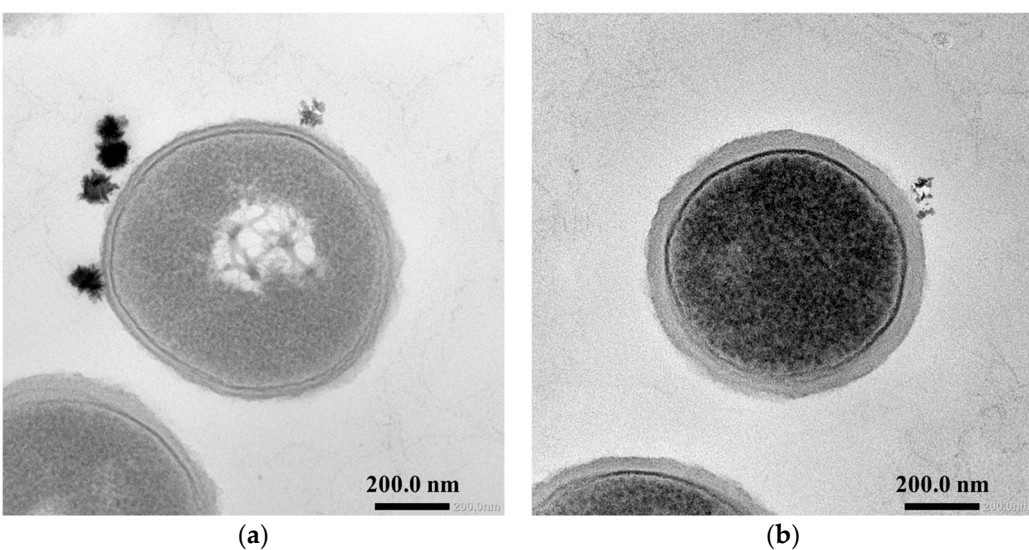

(**a**)                   (**b**)

**Figure 3.** Transmission electron microscopic images of *L. plantarum* JCM1142$^T$. (**a**) early log phase cells, (**b**) early stationary phase cells.

The thickness of the peptidoglycan layer of the cells in each growth phase was measured by ImageJ. Then, the average thickness of the peptidoglycan layers was calculated and compared (Figure 4). The peptidoglycan layer of *L. plantarum* cells in the early log phase was about 17 nm, and that in the early stationary phase was about 45 nm. As a result of the significance test, it was shown that there was a significant difference in the thickness of the peptidoglycan layer between the early log phase and the early stationary phase ($p < 0.001$).

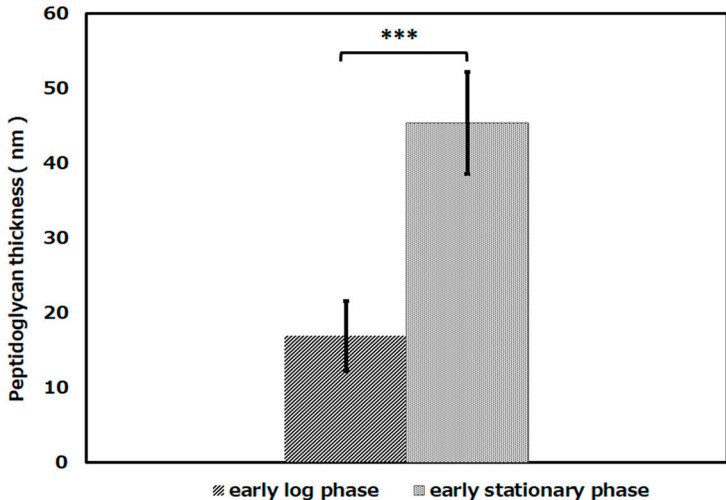

**Figure 4.** The thickness of the peptidoglycan layer of *L. plantarum* JCM1142$^T$ in the early log phase and the early stationary phase. ([early log phase]: $n = 29$, [early stationary phase]: $n = 64$). Error bars indicate the mean value $\pm$ standard deviation of experimental results repeated $n$ times ($p < 0.001$). The triple asterisk (***) indicates a significant difference in the peptidoglycan thickness of early log phase and early stationary phase at $p < 0.001$.

### 2.3. The Survival Rate of E. coli IAM12119$^T$ as A Gram-Negative Bacterium on the Photocatalytic Reaction

2.3.1. The Survival Rate of Spheroplast Cells of *E. coli*

Photocatalytic sterilization was performed on spheroplast cells of *E. coli* IAM12119$^T$ from which the peptidoglycan layer was removed by the enzyme (Lysozyme) and ethylene-diaminetetraacetic acid (EDTA), and the survival rate was evaluated (Figures 5 and 6). The survival rates of spheroplast cells decreased to about 95%, about 63%, and about 57% corresponding to after 1 h, 2 h, and 3 h of the photocatalytic reaction. In negative control experiments ([TiO$_2$ (−), UVA (+)], [TiO$_2$ (+), UVA (−)]), the survival rates after 3 h were about 95% and 99%, respectively.

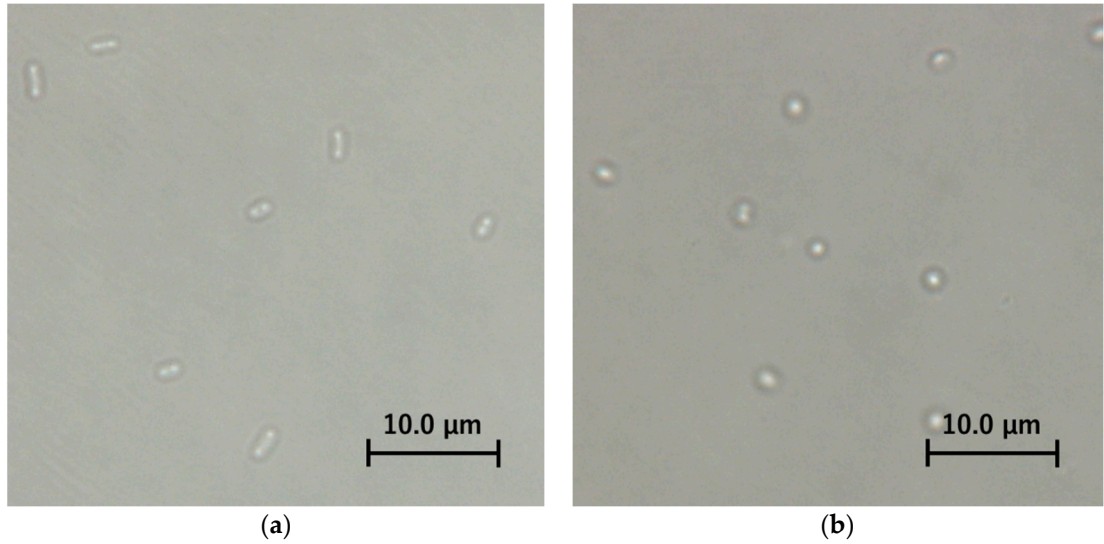

(**a**)          (**b**)

**Figure 5.** Light microscopic bright field images of *E. coli* IAM12119$^T$, (**a**) intact cells, (**b**) spheroplast cells.

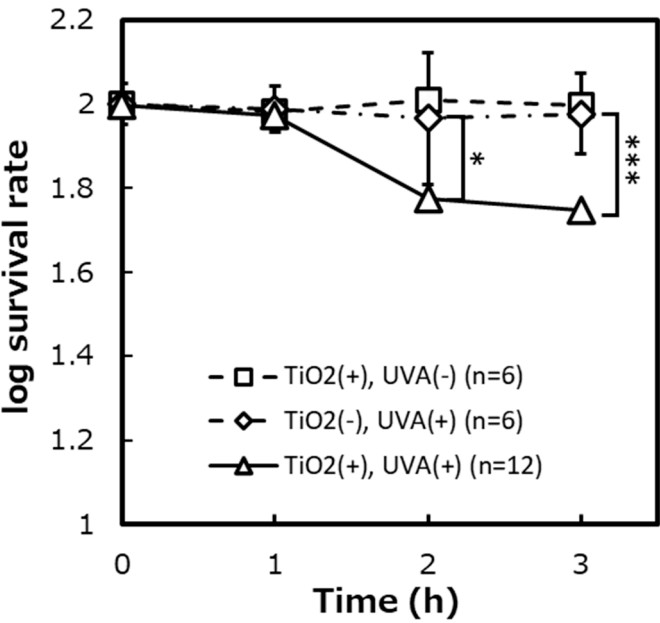

**Figure 6.** The survival rates of spheroplast cells of *E. coli* IAM12119$^T$ on the photocatalytic reaction. ([TiO$_2$ (+), UVA (+)]: $n = 12$, [TiO$_2$ (+), UVA (−)]: $n = 6$, [TiO$_2$ (−), UVA (+)]: $n = 6$). Error bars indicate the mean value ± standard deviation of experimental results repeated $n$ times. ([1 h]: $p < 0.05$, [3 h]: $p < 0.001$). The asterisk (*) and triple asterisk (***) indicate significant differences in the survival rates of [TiO$_2$ (+), UVA (+)] and [TiO$_2$ (−), UVA (+)] at $p < 0.05$ and at $p < 0.001$, respectively.

### 2.3.2. The Comparison of Survival Rates of *E. coli* IAM12119[T] with and without Peptidoglycan Layer

Figure 7 summarizes the survival rates of *E. coli* when the photocatalytic reaction (TiO$_2$ (+), UVA (+)) was performed between the intact cells and the spheroplast cells. The survival rates after 3 h of the photocatalytic reaction were about 41% for the intact cells and about 57% for the spheroplast cells. As a result of the significance test, it was shown that there were significant differences in the survival rates between the intact cells and the spheroplast ($p < 0.001$).

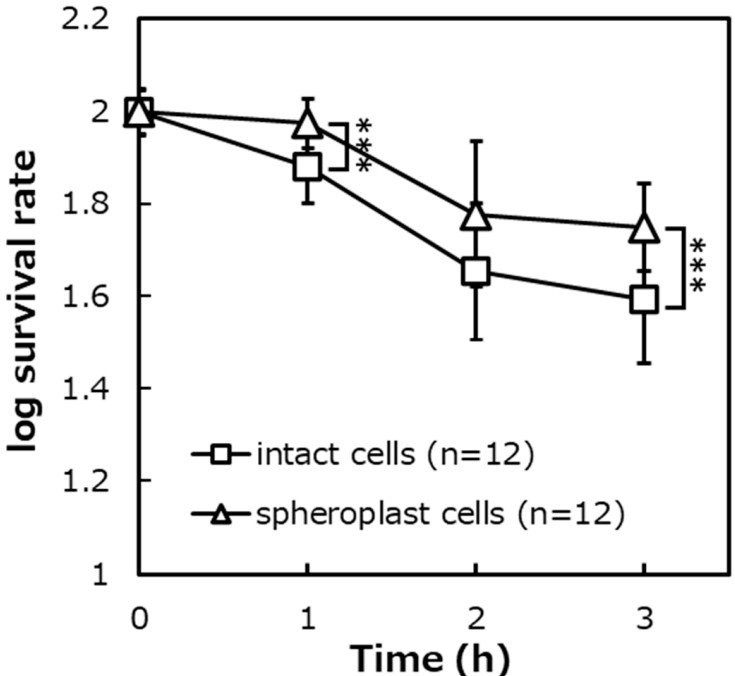

**Figure 7.** The survival rates of *E. coli* intact cells and spheroplast cells on the photocatalytic reaction ([intact cells]: $n = 12$, [spheroplast cells]: $n = 12$). Error bars indicate the mean value ± standard deviation of experimental results repeated *n* times, ([1 h] [3 h]: $p < 0.001$). The triple asterisk (***) indicates a significant difference in the survival rates of intact cells and spheroplast cells at $p < 0.001$.

### 2.3.3. The Effect of Protein Addition on the Photocatalytic Sterilization

To verify the effect of the enzyme added during the preparation of spheroplast cells on the photocatalytic sterilization, albumin without enzyme activity was added to the intact cells and the photocatalytic reaction was carried out (Figure 8).

First, to evaluate the effect of the presence or absence of protein addition on the photocatalytic bactericidal effect, the survival rate of albumin-added intact cells was compared with the survival rate of albumin-free intact cells. The survival rates of the intact cells with albumin were about 75%, about 49%, and about 32%, corresponding to after 1 h, 2 h, and 3 h of the photocatalytic reaction. Additionally, the survival rates of intact cells with albumin were about 77% after 1 h of the photocatalytic reaction, about 47% after 2 h, and about 41% after 3 h. As a result of the significance test, it was shown that there was no significant difference in the survival rates between the presence and absence of protein addition.

Next, to evaluate the effect of the presence or absence of the peptidoglycan layer on the photocatalytic sterilization under protein addition conditions, the survival rates of albumin-added intact cells and the spheroplast cells were compared. The survival rates of the intact cells with albumin were added was about 75%, about 49%, and about 32%, corresponding to after 1 h, 2 h, and 3 h of the photocatalytic reaction. The survival rates of the spheroplast cells were about 95%, about 63%, and about 57% after 1 h, 2 h, and 3 h of the photocatalytic reaction. As a result of the significance test, it was shown that there

was a significant difference in survival rates between albumin-added intact cells and the spheroplast cells after 3 h of the photocatalytic reaction ($p < 0.001$).

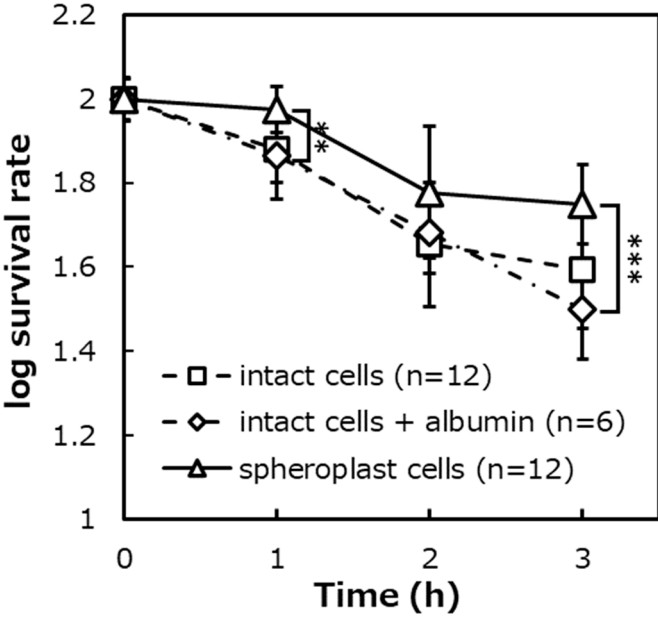

**Figure 8.** The survival rates of protein-added *E. coli* IAM12119[T] intact cells on the photocatalytic reaction, ([intact cells + albumin]: $n = 6$, [intact cells]: $n = 12$, [spheroplast cells]: $n = 12$). Error bars indicate the mean value $\pm$ standard deviation of experimental results repeated $n$ times, ([1 h]: $p < 0.01$, [3 h]: $p < 0.001$). The double asterisk (**) and triple asterisk (***) indicate significant differences in the survival rates of spheroplast cells and intact cells + albumin at $p < 0.01$ and at $p < 0.001$, respectively.

2.3.4. The Effect of EDTA Addition on the Photocatalytic Sterilization

Similar to the experimental objective in Section 2.3.3., to verify the effect of EDTA used during spheroplast cells preparation on the photocatalytic effect, only EDTA was added to the intact cells and the photocatalytic reaction was carried out (Figure 9).

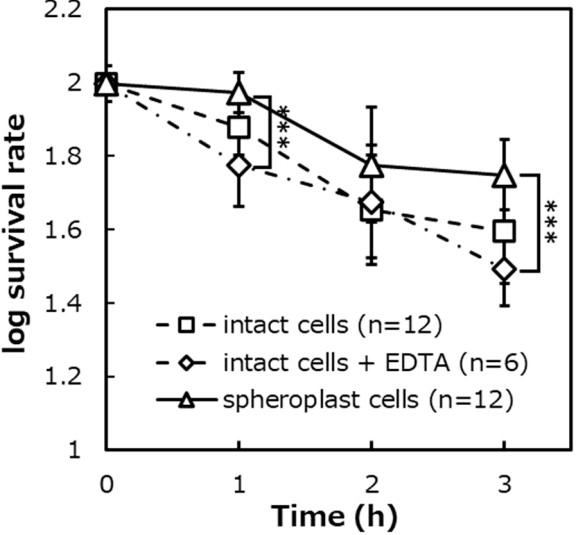

**Figure 9.** The survival rates of EDTA-added *E. coli* IAM12119[T] intact cells on the photocatalytic reaction, ([intact cells + EDTA]: $n = 6$, [intact cells]: $n = 12$, [spheroplast cells]: $n = 12$). Error bars indicate the mean value $\pm$ standard deviation of experimental results repeated $n$ times, ([1 h], [3 h]: $p < 0.001$). The triple asterisk (***) indicates a significant difference in the survival rates of spheroplast cells and intact cells + EDTA at $p < 0.001$.

The survival rates of the intact cells with EDTA were about 61%, about 50%, and about 32% after 1 h, 2 h, and 3 h of the photocatalytic reaction. As a result of the significant difference test, it was shown that there was no significant difference in the survival rates between the intact cells with EDTA and without EDTA.

Moreover, as a result of the significant difference test, it was shown that there were significant differences in the survival rate between the intact cells with EDTA and the spheroplast cells ($p < 0.001$).

### 2.3.5. Evaluation of the Survival Rate of the Spheroplast Cells Added Peptidoglycan

To clarify the cause of the enhancement of the photocatalytic bactericidal effect of the peptidoglycan layer, it was investigated whether the cause of the enhancement was the presence of the peptidoglycan layer in the cell wall or the substance peptidoglycan itself. Therefore, the survival rates of the spheroplast cells added peptidoglycan and the intact cells were compared (Figure 10).

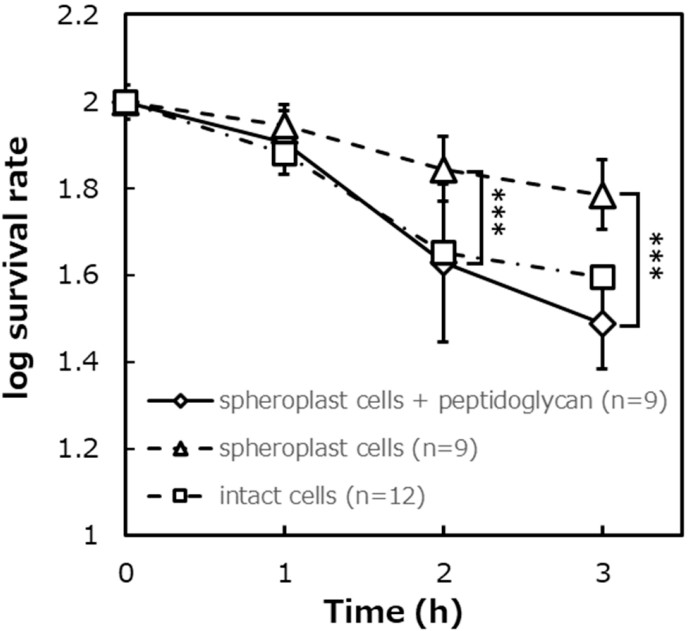

**Figure 10.** The survival rate after the photocatalytic reaction in E. coli IAM12119[T] spheroplast cells added peptidoglycan, ([Spheroplast cells + peptidoglycan]: $n = 9$, [spheroplast cells]: $n = 9$, [intact cells]: $n = 12$). Error bars indicate the mean value $\pm$ standard deviation of experimental results repeated $n$ times, ([2 h], [3 h]: $p < 0.001$). The triple asterisk (***) indicates a significant difference in the survival rates of spheroplast cells and spheroplast cells + peptidoglycan at $p < 0.001$.

Figure 10 shows the results of comparing the survival rates of spheroplast cells to which peptidoglycan was added, spheroplast cells to which peptidoglycan was not added, and intact cells when the photocatalytic reaction was performed. The survival rates of spheroplast cells to which peptidoglycan was not added were about 81% after 1 h of the photocatalytic reaction, about 45% after 2 h, and about 32% after 3 h. The survival rate of spheroplast cells added peptidoglycan was about 89% after 1 h, about 71% after 2 h, and about 62% after 3 h. The survival rate of intact cells was about 77% after 1 h, about 47% after 2 h, and about 41% after 3 h. As a result of a significant difference test between spheroplast cells to which peptidoglycan was added and spheroplast cells to which peptidoglycan preparation was not added, it was shown that there was a significant difference in survival rate between the two conditions after 2 and 3 h ($p < 0.001$). Moreover, as a result of a significant difference test between spheroplast cells to which peptidoglycan was added and intact cells, it was shown that there was no significant difference.

### 2.4. Quantitative Determination of Hydroxyl Radical and Hydrogen Peroxide under Conditions of the Presence and Absence of Peptidoglycan

From the previous study of our laboratory and this study, it was suggested that peptidoglycan promotes the photocatalytic bactericidal effect. However, it is not clear how the peptidoglycan promotes photocatalytic sterilization. Photogenerated reactive oxygen species (ROS) such as hydroxyl radical and hydrogen peroxide are considered to be the major bactericidal agents [22], and we investigated the quantitative determination of hydroxyl radical in the photocatalytic reaction under the condition of different amounts of the peptidoglycan.

The amount of hydroxyl radical generated by the photocatalytic reaction with $TiO_2$ and peptidoglycan was significantly lower than the condition of only $TiO_2$ alone (Figure 11). In the negative control experiments without $TiO_2$, no hydroxyl radical was generated by the peptidoglycan under UVA irradiation.

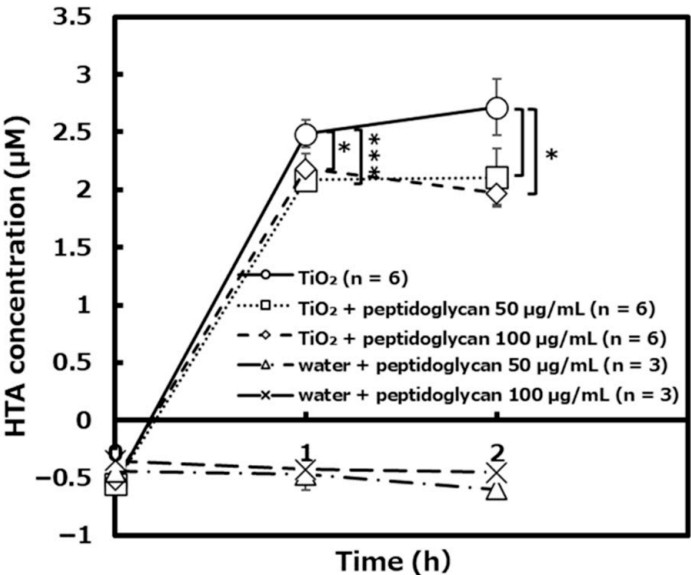

**Figure 11.** The amount of hydroxyl radical generated by $TiO_2$ the photocatalytic reaction in the presence and absence of peptidoglycan ([$TiO_2$]: *n* = 6, [$TiO_2$ + peptidoglycan 50 μg/mL]: *n* = 6, [$TiO_2$ + peptidoglycan 100 μg/mL]: *n* = 6, [water + peptidoglycan 50 μg/mL]: *n* = 3, [water + peptidoglycan 100 μg/mL]: *n* = 3). Error bars indicate the mean value ± standard deviation of experimental results repeated *n* times, ([1 h, $TiO_2$: $TiO_2$ + peptidoglycan 50 μg/mL]: $p < 0.001$, [1 h, $TiO_2$: $TiO_2$ + peptidoglycan 100 μg/mL] [2 h, $TiO_2$: $TiO_2$ + peptidoglycan 50 μg/mL or peptidoglycan 100 μg/mL]: $p < 0.05$). The asterisk (*) and triple asterisk (***) indicate significant differences in HTA concentration of $TiO_2$ and $TiO_2$ + peptidoglycan 50 or 100 μg/mL at $p < 0.05$ and at $p < 0.001$, respectively.

Next, we focused on hydrogen peroxide as active oxygen species that acts on sterilization, and quantified hydrogen peroxide generated when peptidoglycan was photocatalytically treated for 3 h. It was also verified whether the amount of hydrogen peroxide generated changed depending on the amount of peptidoglycan.

The photocatalytic reaction with 50 and 1000 μg/mL of peptidoglycan suspensions for 3 h resulted in significantly higher amounts of hydrogen peroxide generated than in the absence of peptidoglycan (Figure 12). The amount of hydrogen peroxide generated increased in proportion to the amount of peptidoglycan added.

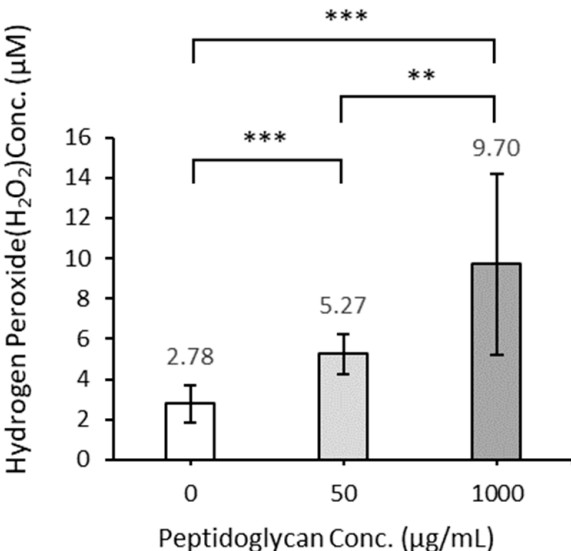

**Figure 12.** The amount of hydrogen peroxide generated by TiO$_2$ photocatalytic reaction in the presence and absence of peptidoglycan ([0 μg/mL]: $n = 21$, [50 μg/mL]: $n = 21$, [1000 μg/mL]: $n = 16$). Error bars indicate the mean value ± standard deviation of experimental results repeated $n$ times, ([0 μg/mL: 50 μg/mL], [0 μg/mL: 1000 μg/mL]: $p < 0.001$, [50 μg/mL: 1000 μg/mL]: $p < 0.01$). The double asterisk (\*\*) and triple asterisk (\*\*\*) indicate significant differences in H$_2$O$_2$ conc. of each peptidoglycan conc. at $p < 0.01$ and at $p < 0.001$, respectively.

## 3. Discussion

A previous study in our laboratory suggested that the peptidoglycan layer enhances the photocatalytic bactericidal effect by the results of photocatalytic experiments using Gram-positive bacterium [27]. However, the quantitative relationship between the thickness of the peptidoglycan layer and the enhancement of photocatalytic bactericidal effect and the photocatalytic bactericidal effect of the peptidoglycan layer using Gram-negative bacterium were not clarified. In this study, we attempted to clarify these factors.

*3.1. The Quantitative Relationship between the Thickness of the Peptidoglycan Layer on a Gram-Positive or Gram-Negative Bacterium and the Enhancement of Photocatalytic Bactericidal Effect*

To verify whether the bactericidal effect of the photocatalytic reaction differs depending on the thickness of the peptidoglycan layer, we hypothesized that the peptidoglycan layer became thicker as the cell grew and performed the photocatalytic reaction on the *L. plantarum* cells in each growth phase. The hypothesis was verified by the observation of transmission electron microscopic observation. The thickness of the peptidoglycan layer of *L. plantarum* cells in the early stationary phase was about three times that in the early log phase (Figures 3 and 4). This result shows that the peptidoglycan layer increase thickens as the cells grow.

From the start of UVA irradiation to 1 h, the survival rate of the early stationary phase of *L. plantarum* cells was lower than that in the early log phase cells (Figure 1). As the cause of significantly reducing the survival rate of early stationary phase cells, it was also thought that the influence of the cell aging or lactic acid produced by *L. plantarum*. The results of the verification showed that *L. plantarum* cells under negative control conditions without photocatalytic reaction rarely died compared to the condition in the photocatalytic reaction even if time passed (Figure 2). From these results, it was suggested that the lactic acid produced by *L. plantarum* during the photocatalytic reaction did not affect the photocatalytic sterilization. In addition, since there is no significant difference between the results of the early log phase and the results of the more advanced early stationary phase, it is considered that cell aging does not affect photocatalytic sterilization. It is suggested that

the peptidoglycan layer promotes the photocatalytic bactericidal effect in the quantitative relationship.

To verify whether the bactericidal effect of the photocatalytic reaction differs depending on the presence or absence of the peptidoglycan layer, the photocatalytic reaction was carried out on intact cells and spheroplast cells of *E. coli*, which is a Gram-negative bacterium. Since the spheroplast cells without the peptidoglycan layer are vulnerable to physical stress when the cells are spread on the agar medium plate, we used LIVE/DEAD® *Bac*Light™ Bacterial Viability which determined survival and dead cell based on the membrane damage instead of the colony counting method. After preparing the spheroplast cells, the cell morphology was observed in the bright field of a light microscope to confirm whether the peptidoglycan layer was removed (Figure 5). The untreated intact cells were short rod-shaped, whereas the enzyme-treated spheroplast cells were spherical, suggesting that the peptidoglycan layer was removed. Before comparing the survival rate of intact cells and spheroplast cells, negative control experiments were performed to confirm that $TiO_2$-coated glass and UVA irradiation did not affect intact cells and spheroplast cells of *E. coli* (Figure 6). As a result, it was shown that $TiO_2$-coated glass and UVA irradiation itself did not show a significant bactericidal effect on intact cells and spheroplast cells.

The survival rates of intact cells and spheroplast cells after the photocatalytic reaction were compared. The result was that the survival rates of intact cells with a peptidoglycan layer were significantly lower than that of spheroplast cells that had lost the peptidoglycan layer (Figure 7). The possibility that the photocatalytic bactericidal effect was suppressed by lysozyme and EDTA added during the preparation of spheroplast cells was denied from the results of verifying experiments (Figures 8 and 9).

### 3.2. Evaluation of Survival Rate of Spheroplast Cells Added Peptidoglycan Preparation

From the above, it was strongly suggested that the peptidoglycan layer enhanced the photocatalytic bactericidal effect. Then, to evaluate whether the state in which the peptidoglycan layer is present in the cell wall is important or simply the component of the peptidoglycan is important, the peptidoglycan was added to spheroplast cells and the photocatalytic reaction was carried out (Figure 10). Negative control experiments were performed to show that the addition of peptidoglycan did not negatively affect spheroplast cells. As a result, the addition of the peptidoglycan did not show a significant negative effect on spheroplast cells.

Then, the survival rates of the spheroplast cells added peptidoglycan, and the spheroplast cells without peptidoglycan were compared. Three hours after the photocatalytic reaction, the survival rate of the spheroplast cells added peptidoglycan was significantly lower than that of the spheroplast cells without peptidoglycan. Furthermore, there was no significant difference in the survival rate of spheroplast cells added peptidoglycan and intact cells. This suggests that for peptidoglycan to affect the bactericidal effect, the peptidoglycan layer does not need to be present in the cell as the cell wall, and the component of peptidoglycan is important.

### 3.3. Quantification of Hydroxyl Radical and Hydrogen Peroxide

From the above results, it is suggested that the peptidoglycan layer promotes the photocatalytic bactericidal effect in proportion. However, since it is not clear how peptidoglycan enhances photocatalytic sterilization, it was quantitatively confirmed whether the addition of peptidoglycan to the photocatalytic reaction increased the amount of hydroxyl radical and hydrogen peroxide. As a result, when peptidoglycan was added to $TiO_2$, the amount of hydroxyl radical produced was lower than that of $TiO_2$ alone (Figure 11). It was also shown that the amount of hydrogen peroxide generated during the photocatalytic reaction increased in proportion to the concentration of peptidoglycan (Figure 12). These results suggest that peptidoglycan consumed the hydroxyl radical produced by the photocatalytic reaction to produce hydrogen peroxide. The enhancing effect of photocatalytic sterilization by peptidoglycan is thought to be achieved by the production of ROS such

as hydrogen peroxide instead of hydroxyl radical. We think that the decrease in hydroxyl radicals is due to peptidoglycan, but the relationship between that reaction and the reaction that leads to the increase in hydrogen peroxide cannot be speculated at this time. We are currently considering ways to find out more.

To summarize this study, like the study of the Gram-positive bacterium in our previous study, the Gram-negative bacterium was shown to be a high bactericidal effect in the presence of the peptidoglycan layer. It was also revealed that the thicker the peptidoglycan layer, the higher the photocatalytic bactericidal effect. It was shown that the cause of the photocatalytic sterilization enhancing effect of peptidoglycan may be the production of new ROS such as hydrogen peroxide by peptidoglycan.

## 4. Materials and Methods

### 4.1. Bacterial Strains

*L. plantarum* JCM1149[T] and *E. coli* IAM12119[T] were used for the photocatalytic bactericidal experiments. These strains were obtained from Japan Collection of Microorganisms, RIKEN BioResource Research Center, Japan and Institute of Applied Microbiology, The University of Tokyo, Japan.

### 4.2. Preparation of Bacterial Suspension

#### 4.2.1. *L. plantarum* Cells

*L. plantarum* JCM1149[T] cells were statically precultured in de Man, Rogosa, and Sharpe (MRS) broth (Becton, Dickinson and Company, Franklin Lakes, NJ, USA) at 30 °C for 24 h. In the main culture, the bacterial suspension was inoculated into the new MRS broth, and static-cultured in an incubator at 30 °C until each growth phase. The growth phase begins in the lag phase, goes through the log phase, the stationary phase, and then to the death phase. Each phase is divided into three, early, middle, and late. Each growth phase was divided with reference to *L. plantarum* growth curve that we measured and created. The cells were washed three times with $1 \times$ PBS, re-suspended in the new $1 \times$ PBS to OD (Optical Density) 660 nm of 1.00. This solution was used as a bacterial suspension of *L. plantarum* cells.

#### 4.2.2. *E. coli* Cells and Spheroplast Cells

*E. coli* IAM12119[T] cells were precultured with shaking in nutrient broth (Nissui Pharmaceutical Co., Ltd., Tokyo, Japan) in an incubator (Tokyo Rika Kikai, Tokyo, Japan) at 37 °C for 18 h. In the main culture, the bacterial suspension was inoculated into the new nutrient broth, and shake-cultured in an incubator at 37 °C for 2.5 h until the early log phase. Furthermore, $1 \times$ PBS was used as a bacterial suspension of *E. coli* intact cell.

The cells cultured in the same manner as above were washed 3 times with 0.03 M Tris-HCl buffer containing 20% sucrose. The cells were re-suspended in 0.03 M Tris-HCl buffer containing 20% sucrose to OD 660 nm of 0.50. Lysozyme solution was added to the bacterial suspension to a final concentration of 1 mg/mL, and 0.1 M EDTA solution was also added. After mixing by inversion, the mixture stood at 30 °C for 1 h. Then, the mixture was centrifuged and resuspended in 10 mM PBS buffer containing 0.2% $MgCl_2$. This solution was used as a bacterial suspension of spheroplast cells. The concentration of each reagent was determined considering the optimal conditions for the preparation of spheroplast cells.

### 4.3. Photocatalytic Experimental Device

In this study, the $TiO_2$-coated glass was used to investigate the photocatalytic bactericidal effect, which does not need into consideration in nanoparticle toxicity. The $TiO_2$-coated glass was prepared as follows. A flat glass plate ($50 \times 50 \times 2$ mm$^3$, Tokyo Glass Instruments, Tokyo, Japan) was washed with 0.5% NaOH solution and water. An amount of 1 mL of $TiO_2$ coating solution (TKC-304, Tayca Co., Ltd., Osaka, Japan) was dropped

onto the washed glass plate, spin-coated (3000 rpm, 5 s) using a spin coater (MS-A200, Mikasa Co., Ltd., Tokyo, Japan), and dried at room temperature.

The photocatalytic reaction was carried out by constructing an evaluation device according to Japanese Industrial Standards (JIS R-1702) described in the previous paper. All parts of the evaluation device were washed with 70% ethanol and exposed to UVC overnight. Filter paper moistened with sterilized water, a plastic stage, and $TiO_2$-coated glass plate were placed in order from the bottom of a petri dish.

### 4.4. The Photocatalytic Reaction

In the experiments with *L. plantarum*, 100 μL of bacterial suspension was dropped onto the $TiO_2$-coated glass plate. An OHP film ($45 \times 45$ mm$^2$, KOKUYO Co., Ltd., Osaka, Japan) was placed on it, and the entire petri dish was covered with a wrap. Using the black light FL15BL-B (Panasonic, Osaka, Japan) as light source, the irradiation intensity of UVA was adjusted to 0.25 mW/cm$^2$ as described in the previous paper, and the experimental device was irradiated with UVA for a maximum of 2 h. This irradiation intensity corresponds to the intensity of the sunlight shining into the window during the daytime. After the photocatalytic reaction, the $TiO_2$-coated glass plate with bacterial cells and 10 mL of PBS buffer were placed in a sterile plastic bag and washed by rubbing to recover the bacterial suspension. The serially diluted bacterial suspension was spread on the Nutrient agar (Nissui Pharmaceutical Co., Ltd., Tokyo, Japan), incubated at 37 °C overnight, and the number of surviving cells was determined.

In the experiments with *E. coli*, 0.3 μL of LIVE/DEAD® *Bac*Light™ Bacterial Viability Kit solution (Molecular Probes, Eugene, OR, USA) was added to 100 μL of bacterial suspension, and 10 μL of this bacterial suspension was dropped to $TiO_2$-coated glass plate. A cover glass ($22 \times 22$ mm$^2$) was placed thereon, and the entire petri dish was covered with a wrap. The irradiation intensity of UVA was adjusted to 0.25 mW/cm$^2$ as described in the previous paper. The photocatalytic reaction was performed for a maximum of 3 h as the fluorescent pigments of the LIVE/DEAD® *Bac*Light™ Bacterial Viability Kit faded with prolonged UVA irradiation. After the photocatalytic reaction, the fluorescent cell was observed using an all-in-one fluorescence microscope BZ-8100 (KEYENCE) to calculate the cell membrane non-damage rate, that is, the survival rate.

### 4.5. Evaluation of the Survival Rate of the Spheroplast Cells Added Peptidoglycan

In the experiments of peptidoglycan addition, the peptidoglycan prepared from *Bacillus subtilis* (Sigma-Aldrich, St. Louis, MO, USA) was added to the bacterial suspension of spheroplast cells to a final concentration of 50 μg/mL.

### 4.6. Transmission Electron Microscopic Observation of L. plantarum Cells in Each Growth Phase

To clarify that the peptidoglycan layer of *L. plantarum* cell thickens with cells growth, we observed the early log phase cells and the early stationary phase cells with a transmission electron microscope. *L. plantarum* cells cultured up to the above two growth phases were collected and washed three times with PBS buffer. The cells were fixed in 2% glutaraldehyde (2% glutaraldehyde, 0.1 M $NaH_2PO_4$, 0.1 M $Na_2HPO_4$) at 4 °C for 1.5 h, washed five times with 0.1 M phosphate buffer (0.1 M sucrose, 0.1 M $NaH_2PO_4$, 0.1 M $Na_2HPO_4$) at 4 °C for 30 min, and postfixed with 1.0% $OsO_4$ (1.0% $OsO_4$, 0.1 M sucrose, 0.1 M $NaH_2PO_4$, 0.1 M $Na_2HPO_4$) at 4 °C for 1.5 h. Fixed cells were embedded in 2% agarose (Agarose S, FUJIFILM Wako Pure Chemical Co., Ltd., Osaka, Japan) and cut into 1 mm$^3$ cube. The cubes were dehydrated through 50, 70, 80, 90, and 95% ethanol solutions at 4 °C for 10 min at each dilution, followed by dehydration with 99.5% ethanol twice for 30 min. After dehydration, we discarded half of 99.5% ethanol and added QY-1 (*n*-butyl glycidyl ether, FUJIFILM Wako Pure Chemical Co., Ltd., Osaka, Japan) to replace ethanol in the cubes at room temperature for 5 min. Subsequently, all of the reagents were discarded, and an equal amount of QY-1 was added to replace ethanol completely in the cubes at room temperature for 10 min. After half of the reagents were discarded,

an equal amount of epoxy resin (Epon 812, Ouken Trading Co., Ltd., Tokyo, Japan) was added and penetrated with shaking (80 rpm, room temperature) for 2 h. All amounts of the reagents were discarded, followed by completely penetration twice with new epoxy resin for 1 h (120 rpm). New epoxy resin was poured into the mold and the cube was embedded in the tip. Subsequently, the epoxy resin containing the cube was polymerized at 60 °C for 24 h. Ultra-thin section preparation, staining, and observation were performed at the contract-based analysis company (JEOL Ltd., Tokyo, Japan).

### 4.7. Quantitative Determination of Hydroxyl Radical

In order to quantify the hydroxyl radical generated during the photocatalytic reaction, we used chemical dosimetry based on terephthalic acid (TA) [28]. The hydroxyl radical change terephthalic acid to 2-hydroxyterephthalic acid (HTA), which can be detected by fluorescence measurement. An amount of 100 μL of $TiO_2$ coating solution (Tersus EN, Shin-Etsu Astech Co., Ltd., Tokyo, Japan) and 100 μL of 2 mM TA solution (FUJIFILM Wako Pure Chemical Co., Ltd., Osaka, Japan) were added to the 96well plate. TA solution was prepared by dissolving TA in the distilled water containing 5 mM NaOH (FUJIFILM Wako Pure Chemical Co., Ltd., Osaka, Japan). The peptidoglycan prepared from *Bacillus subtilis* (Sigma-Aldrich, St. Louis, MO, USA) was added to the well to a final concentration of 50 and 100 μg/mL. After covering the 96 well plate with a wrap, the plate was irradiated with 0.25 $mW/cm^2$ of UVA for a maximum of 2 h. After irradiation, the fluorescence of HTA was measured with FluoroCount (PACKARD, Detroit, MI, USA).

### 4.8. Quantitative Determination of Hydrogen Peroxide

Quantitative determination of hydrogen peroxide was performed using Hydrogen Peroxide Colorimetric Detection Kit (Arbor Assays LLC, Ann Arbor, MI, USA). An amount of 100 μL of 50, and 1000 ng/mL peptidoglycan suspension or sterilized water was dropped to $TiO_2$-coated glass in the evaluation device. The irradiation intensity of UVA was adjusted to 0.25 $mW/cm^2$, and UVA irradiation to the evaluation device was performed for 3 h. After irradiation, the solution on the glass was collected, and 50 μL of the collected solution was added to the 96 well plate. After reacting with Hydrogen Peroxide Colorimetric Detection Kit solution for 15 min, the absorbance was measured with SpectraMax ABS Plus (Molecular Devices, San Jose, CA, USA).

**Author Contributions:** Investigation—study with *L. plantarum* and hydroxyl radical, A.S.; Investigation—study with *E. coli* and hydrogen peroxide, K.S.; writing—original draft preparation A.S. and K.S.; writing—review and editing, T.S.; supervision, T.S.; project administration, T.S. All authors have read and agreed to the published version of the manuscript.

**Funding:** This research received no external funding.

**Institutional Review Board Statement:** Not applicable.

**Informed Consent Statement:** Not applicable.

**Data Availability Statement:** Data is contained within the article.

**Acknowledgments:** We are grateful Distinguished Akira Fujishima and Chiaki Terashima for providing us a place to research and useful suggestion. We also thank Tayca Co., Ltd. for providing us $TiO_2$ coating solution (TKC-304).

**Conflicts of Interest:** The authors declare no conflict of interest.

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
