# Peer review of "Analyses of the Effect of Peptidoglycan on Photocatalytic Bactericidal Activity Using Different Growth Phases Cells of Gram-Positive Bacterium and Spheroplast Cells of Gram-Negative Bacterium"

_catalysts, doi:10.3390/catal11020147_

Round 1
Reviewer 1 Report
Authors did a good job of presenting their work.
Author Response
Dear Reviewer1
Thank you very much for reviewing.
We are thankful for the time and energy you expended.
Due to a typographical error in the values below 0 on the vertical axis, Figure 1 has been replaced.
Reviewer 2 Report
The authors report a series of studies to investigate the role of peptidoglycan on the bactericidal activity of photocatalytic systems. For this purpose, they used different bacteria types and evaluated the generation of hydroxyl radicals and hydrogen peroxide. Overall, the results are interesting but there is a strong similarity with the previous work reported by the same group (Biocontrol Science, 2020, Vol. 25, No. 3, 167—171). The title and the findings are quite similar. Therefore, I cannot recommend this article for publication in its present form.
Additional comments:
1) The terms “early log phase” and “early stationary phase” should be explained in the manuscript.
2) Figure 1 is confusing. The reaction hours and photocatalyst type should be added in the description of the Figure.
3) TEM images of Escherichia coli cells should be added to corroborate the absence of the peptidoglycan layer.
Author Response
Dear Reviewer2
Thank you very much for providing important comments. We are thankful for the time and energy you expended. Our responses to your comments are as follow:
- The authors report a series of studies to investigate the role of peptidoglycan on the bactericidal activity of photocatalytic systems. For this purpose, they used different bacteria types and evaluated the generation of hydroxyl radicals and hydrogen peroxide. Overall, the results are interesting but there is a strong similarity with the previous work reported by the same group (Biocontrol Science, 2020, Vol. 25, No. 3, 167—171). The title and the findings are quite similar. Therefore, I cannot recommend this article for publication in its present form.
ï½¥RESPONSE: Thank you very much for providing important comments. The previous study (Biocontrol Science, 2020, Vol. 25, No. 3, 167—171) has suggested that gram-positive bacterial peptidoglycan may promote photocatalytic bactericidal effects. In this study, it was newly revealed that the same possibility was suggested in Gram-negative bacteria and showed that our suggestions apply to bacteria in general. Furthermore, it was newly revealed that the promotion of photocatalytic bactericidal effect by peptidoglycan depends on the amount of peptidoglycan. The above two points are the significance of this research, and we believe that it is possible to differentiate it from the previous study.
- The terms “early log phase” and “early stationary phase” should be explained in the manuscript.
ï½¥RESPONSE: Thank you for your suggestion. The terms “early log phase” and “early stationary phase” are defined from the bacterial growth curve. The growth phase begins in the lag phase, goes through the log phase, the stationary phase, and then to the death phase. And each phase is divided into three, early, middle, and late. We added this description to the manuscript (p. 12-13, lines 358-361).
- Figure 1 is confusing. The reaction hours and photocatalyst type should be added in the description of the Figure.
ï½¥RESPONSE: Thank you for providing these insights. The reaction hours and photocatalyst type were added in the description of Figure 1.
- TEM images of Escherichia coli cells should be added to corroborate the absence of the peptidoglycan layer.
ï½¥RESPONSE: We think it is necessary to add TEM images of Escherichia coli cells, but since it can be shown that peptidoglycan has disappeared by an optical microscope, We have added that photo this time. Escherichia coli, with a peptidoglycan layer, exhibits a short rod-shaped morphology. When the peptidoglycan layer disappeared or became considerably thin due to the enzyme, the cell cannot maintain its morphology and deforms into a spherical shape. Therefore, we have added comparative images of intact cells and spheroplast cells to show this (p. 5, Figure 5).
Due to a typographical error in the values below 0 on the vertical axis, Figure 1 has been replaced.
Again, thank you for giving us the opportunity to strengthen our manuscript with your valuable comments and queries. We have worked to incorporate your feedback and hope that these revisions persuade you to accept our submission.
Reviewer 3 Report
This paper presents a high potential for publication since its significance is of high interest.
In the application of Advanced Oxidation Processes, such as photocatalysis, there are some studies that show higher bactericidal efficiency on Gram-negative in front of Gram-positive bacteria, but these studies do not deeply study the effect of the peptidoglycan layer (see for instance references below that should be cited in the introduction to make a proper state of the art). This study aims to assess this layer specifically, so it is a good study that can help to clearly elucidate the role of this biological factor in the assessment of some Advanced Oxidation Processes.
However, the study cannot be accepted in its present form. Low quality of redaction, vague language, low readability, and methods poorly described results in a low scientific quality paper that should not be accepted by a high-quality journal as Catalysts is.
Nonetheless, according to my previous comments, I strongly suggest to improve the paper and resubmit it in the future.
References:
Appl. Catal. B Environ. (2010). 98, 27–38. https://doi.org/10.1016/j.apcatb.2010.05.001
Photochem. Photobiol. Sci. (2019). 18, 878–883. https://doi.org/10.1039/C8PP00304A
J. Photochem. Photobiol. A Chem. (2007). 186, 335–341. https://doi.org/10.1016/j.jphotochem.2006.09.002
Appl. Catal. B Environ. (2010). 100, 212–220. https://doi.org/10.1016/j.apcatb.2010.07.034
Author Response
Dear Reviewer3
Thank you very much for providing important comments. We are thankful for the time and energy you expended.
The manuscript has been revised to reflect your suggestions, citing references that have been kindly suggested. We have worked to incorporate your feedback and hope that these revisions persuade you to accept our submission.
Due to a typographical error in the values below 0 on the vertical axis, Figure 1 has been replaced.
Reviewer 4 Report
The present ms describes the effect of peptidoglycan on the bactericidal activity of TiO2 photocatalysis. While the organization of the experimental activity and the ms is really good, I find that the style and language should be markedly improved prior to publication.
Overall, I think that this work is the result of a robust experimental investigation and I have no comments on these aspects. I only recommend to add suitable introductory paragraphs for non-experts to clearly describe:
- the evolution of the peptidoglycan layer with cell growth (also considering different cell types);
- how a photocatalyst works, clearly highlighting why hydroxyl radicals and H2O2 are important in the evaluation of the photocatalytic activity.
In addition, I have a comment related to an argument from the Discussion, section 3.1. It is not clear to me how the Atuhors determined that the tendency of cells to die " ... was not affected by cell aging or lactic acid produced." (see lines 272-273). Please add relevant comments.
Can the Authors speculate on the chemical effect of peptidoglycan in decreasing the amount of hydroxyl radical and increasing that of H2O2? Please comment.
Some paragraphs from the template of the journal are present in the ms and should be removed, accordingly. See lines: 96, 430-441.
Author Response
Dear Reviewer4
Thank you very much for providing important comments. We are thankful for the time and energy you expended. Our responses to your comments are as follow:
- I only recommend to add suitable introductory paragraphs for non-experts to clearly describe:
1-1: the evolution of the peptidoglycan layer with cell growth (also considering different cell types);
ï½¥RESPONSE: Thank you for your suggestion. This study showed that the peptidoglycan layer in the early stationary phase, which grew more than the early log phase, was thicker. We added this description to the manuscript (p. 12-13, lines 358-361).
1-2: how a photocatalyst works, clearly highlighting why hydroxyl radicals and H2O2 are important in the evaluation of the photocatalytic activity.
ï½¥RESPONSE: We agree to highlight why hydroxyl radicals and H2O2 are important in the evaluation of the photocatalytic activity. Therefore, text to highlight was added to the manuscript (p. 9, lines 228-229).
- In addition, I have a comment related to an argument from the Discussion, section 3.1. It is not clear to me how the Authors determined that the tendency of cells to die " ... was not affected by cell aging or lactic acid produced." (see lines 272-273). Please add relevant comments.
ï½¥RESPONSE: Thank you very much for your important comments. Based on the results shown in Fig. 2, it was decided that lactic acid and cell aging had no effect. L. plantarum produces lactic acid as it grows. It was denied that the lactic acid produced by L. plantarum could kill L. plantarum as it was shown that L. plantarum rarely dies over time under negative control conditions without photocatalytic reaction. In addition, there is no significant difference between the results of the early log phase and the early stationary phase in which the cells have grown more advanced, eliminating the possibility that the cells died from aging. The above sentences have been added to the text (p. 11, lines 280-286).
- Can the Authors speculate on the chemical effect of peptidoglycan in decreasing the amount of hydroxyl radical and increasing that of H2O2? Please comment.
ï½¥RESPONSE: You have raised an important question. We think that the decrease in hydroxyl radicals is due to peptidoglycan, but the relationship between that reaction and the reaction that leads to the increase in hydrogen peroxide cannot be speculated at this time. We are currently considering ways to find out more.
Due to a typographical error in the values below 0 on the vertical axis, Figure 1 has been replaced.
Again, thank you for giving us the opportunity to strengthen our manuscript with your valuable comments and queries. We have worked to incorporate your feedback and hope that these revisions persuade you to accept our submission.
Round 2
Reviewer 2 Report
The authors have successfully addressed the suggested comments/changes. However, in order to clearly differentiate this work from the previous one (Biocontrol Science, 2020, Vol. 25, No. 3, 167—171), the title of this manuscript should be changed to illustrate the main aim of this work.
Author Response
Dear Reviewer 2
Thank you very much for providing important comments. We are thankful for the time and energy you expended.
The title of the manuscript has been revised to reflect your suggestions. We have worked to incorporate your feedback and hope that these revisions persuade you to accept our submission.